# The Exon 3-Deleted Growth Hormone Receptor (d3GHR) Polymorphism—A Favorable Backdoor Mechanism for the GHR Function

**DOI:** 10.3390/ijms241813908

**Published:** 2023-09-10

**Authors:** Ghadeer Falah, Lital Sharvit, Gil Atzmon

**Affiliations:** 1Faculty of Natural Sciences, University of Haifa, Haifa 3498838, Israel; ghadeer.fa3@gmail.com (G.F.); lsharvit@univ.haifa.ac.il (L.S.); 2Departments of Medicine and Genetics, Albert Einstein College of Medicine, Bronx, NY 10461, USA

**Keywords:** growth hormone receptor, human growth hormone, deletion of exon 3, polymorphism, growth and development, hormone deficiency

## Abstract

Growth hormone (GH) is a peptide hormone that plays a crucial role in controlling growth, development, and lifespan. Molecular regulation of GH is accomplished via the *GH receptor* (*GHR*), which is the main factor influencing human development and is essential to optimal functioning of the GH/IGF-I axis. Two GHR isoforms have been studied, according to the presence (flGHR) or absence (d3GHR) of exon 3. The d3GHR isoform, which lacks exon 3 has recently been related to longevity; individuals carrying this isoform have higher receptor activity, improved signal transduction, and alterations in the treatment response and efficacy compared with those carrying the wild type (WT) isoform (flGHR). Further, studies performed in patients with acromegaly, Prader–Willi syndrome, Turner syndrome, small for gestational age (SGA), and growth hormone deficiency (GHD) suggested that the d3GHR isoform may have an impact on the relationship between GH and IGF-I levels, height, weight, BMI, and other variables. Other research, however, revealed inconsistent results, which might have been caused by confounding factors, including limited sample sizes and different experimental methods. In this review, we lay out the complexity of the GHR isoforms and provide an overview of the major pharmacogenetic research conducted on this ongoing and unresolved subject.

## 1. Introduction

Growth hormone (GH) is a peptide hormone that plays a crucial role in controlling growth, development, and lifespan [1]. Humans have two GHR isoforms that differ in exon 3 retention or exclusion during splicing: a full-length isoform and a d3GHR isoform that lacks exon 3 [2]. This isoform is characterized by increased receptor activity, enhanced signal transduction and altered hormone binding [3,4]. Several studies investigated how this polymorphism might affect growth [5,6,7,8]. Additionally, the clinical and biochemical phenotype of patients with GH abundance or deficiency, such as those with GHD, PWS, acromegaly, Turner syndrome, and small for gestational age (SGA), were studied, along with their responsiveness to particular treatments. Several studies discovered that, depending on the syndrome under investigation, the presence or absence of the d3GHR polymorphism has a positive or negative impact, respectively, on GH concentrations following treatment [9,10,11,12]. In a study of extremely long-lived individuals, d3GHR carriers demonstrated an approximately 10-year lifespan increase and d3GHR homozygotes across ages were an inch taller than WT individuals [13], and they have a significantly higher rate of postnatal catch-up growth [5]. In addition, in acromegaly patients, the d3GHR isoform was associated with a better clinical response to pegvisomant therapy [11]. This review presents information about the functional characteristics of the GHR with a particular emphasis on the d3GHR genotypes and their impact in numerous syndromes related to an excess or lack of GH. It also provides an overview of the major pharmacogenetic studies conducted on these ongoing and unsolved issues.

## 2. GHR Function and Structure

Growth hormone (GH) is a peptide hormone that is pulsatilely released from the somatotroph cells of the anterior pituitary. Growth hormone-releasing hormone (GHRH; positive regulation) and somatostatin (negative regulation) are the main hypothalamic hormones that control the release of GH [14,15]. GH performs an essential function in regulating lifespan, growth, and development [1], the metabolic function of GH affects protein, carbohydrate, and lipid metabolism [15]. The GHR is mediated by the *GHR*, which is located on the short arm of chromosome 5, spanning at least 87 kb on the 5p13.1-p12 chromosome, consists of 10 exons, of which exons 2–10 code for the receptor protein and exon 1 contribute to the 5 untranslated region. Exon 2 codes for the signal peptide (encodes the last 11 bp of the 5′-UTR, an 18-amino-acid signal peptide, and the first 6 amino acids of GHR’s extracellular domain), exons 3–7 for the extracellular domain (246 amino acids), exon 8 for the transmembrane domain (24 amino acids), and exons 9–10 for the intracellular domain (350 amino acids) [3,16,17,18,19]. Additionally, the intracellular domain contains two conserved regions known as Box 1 and Box 2 [20,21]. “Box1” is a proline-rich segment, and “Box2” is a hydrophobic segment [21,22]. They are important for the receptor’s signaling transduction [20,21] and are necessary for activation of both a tyrosine phosphorylation of Jak2 and a GH-stimulated binding activity (GHSF) [23] (Figure 1).

The GHR initiates many intracellular signaling pathways that lead to glucose metabolism alterations, modulation of cell proliferation genes, and generation of insulin-like growth factor (IGF)-1 [15]. When GH binds to its receptor, JAK2 and the receptor are phosphorylated, which initiates signaling via multiple pathways as shown in Figure 2. Receptor dimerization triggers a series of subsequent protein phosphorylations. STAT1, STAT3, and STAT5 are among the STAT proteins that GH can activate. The JAK-STAT pathway is considered as the main pathway by which GH affects gene transcription [24]. Stats 5a and 5b are involved in the synthesis of several GH-sensitive genes, including insulin-like growth factor 1 (IGF-1) and the acid labile subunit (ALS), a vital part of the IGF binding protein complex (IGF1-IGFBP 3-ALS) [25]. Tyrosyl phosphorylation of IRS proteins would be triggered by JAK2 activation, which promotes the recruitment of PI3K, regulating glucose transport and probably other cellular responses [26]. Apoptosis and autophagy are inhibited by the upregulation of Akt, which promotes the phosphorylation of target genes and proteins. This might be because PI3K/Akt triggering causes the NF-kB system to become overactive [27]. Increased the expression of anti-apoptotic proteins such as Mcl-1 has been linked to the capacity of the PI3K pathway to mediate immunotherapy resistance [28,29]. The SHC/MAP kinase pathway is an additional pathway. Mitogen-activated protein kinases (MAPK) and the phosphoinositide-3-kinase/Akt signaling pathways are typically activated by SHC proteins to carry out their actions [30,31]. The pathway consisting of Grb2-SOS-Ras-Raf-MEK-ERK1,2 is activated by SHC adapter proteins [26]. The GHR also activates the Src family kinase signaling pathway independently of JAK2, and leads to the activation of RAS and the beginning of the ERK1/2 signaling pathway, which controls cytosolic targets and gene transcription [32,33] (Figure 2).

Growth hormone signaling is thought to be a key aging regulator [34,35]. Lack of GH signaling significantly expands lifespan, slows the aging process, increases the health span, reduces body size, delays maturation, and increases life expectancy [35]. For instance, a study found that mice homozygous for the GHR disruption lived significantly longer than WT mice; the disruption resulted in a 55% and 38% lifespan increase in males and females, respectively [36]. The loss of receptor activity is mainly due to mutations in the extracellular domain that result in receptor insensitivity or Laron syndrome. However, excessive receptor stimulation by growth hormone leads to gigantism, adult acromegaly, and cancer [37].

Growth hormone receptor deficiency (GHRD) in humans is brought on by *GHR* mutations that result in low levels of both insulin and insulin-like growth factor 1 (IGF-1) [38]; as a result, the circulating levels of IGF-1 affect brain structure and function during development and aging [39]. By focusing on people with GHRD, numerous studies sought to clarify and analyze the role of the GHR. A recent study of 13 people with GHRD and 12 unaffected relatives found that the GHRD group showed trends toward larger dentate gyrus and CA1 regions of the hippocampus as well as larger surface areas in several frontal and cingulate regions. Additionally, the GHRD group exhibited improved task-related activation and cognitive performance in the frontal, parietal, and hippocampal regions when compared with the controls [40]. By contrast, a different study in the Ecuadorian population revealed that people with GHRD are not significantly different in terms of their intellectual ability, and the production of IGF-1 that is induced by GH is not necessary for postnatal intellectual development or for normal brain growth in utero [41]. Yet, a study that looked at 10 patients with the Laron syndrome revealed parenchymal loss to varying degrees and below-average intelligence [42]. Other studies showed that individuals with GHRD had no diabetes, and only one had a non-lethal malignancy, compared to 17% with cancer and 5% with diabetes in the controls [43]. In the light of the various results discussed above, more research is required to better understand and demonstrate how these conserved genes contribute to aging and longevity.

## 3. GHR Isoforms

In 2004, Dos Santos et al. reported on two isoforms of a common GHR, the WT, which contains exon 3 (flGHR), and the mutated form, which lacks it (d3GHR) [2]. Exon 3 encodes a portion of the extracellular domain of the receptor and its absence or presence results in differential GH-binding abilities [44,45]. Since then, a wide range of studies have examined how this GHR polymorphism affects people with various conditions. This article compares the effects of both isoforms in various disorders and unbiasedly assesses these studies.

## 4. Deletion of Exon 3 in GHR (d3GHR)

One of the GHR isoforms is known as “d3GHR” and is encoded by a transcript variant that is missing exon 3. Exon 3 is coding for a portion of the extracellular domain, and its deletion results in a loss of 66 base pairs (bp) that code for 22 amino acids (5′ TAAILSRAPWSLQSVNPGLKT 3′) in the N-terminal domain (residues 7–28) [17,45,46]. By alternative splicing of exon 3, the *GHR* produces two distinct mRNAs in some cell types and placenta, one of which codes for a flGHR and the other for a d3GHR isoform [46]. It is possible that this polymorphism may be connected to polygenic events given the relatively high frequency of homozygous d3GHR expression [47]. It has been suggested that this alternative splice event is tissue-, developmentally, and/or individual-specific [16]. Another possible explanation for the occurrence of these isoforms is that they are a result of an intrachromosomal recombination event between two related primate-specific retro-elements (DNA sequences derived from a retrovirus) flanking exon 3, which took place late in primate evolution [16]. These sequences underwent recombination during cell division, which resulted in a 2.7 kb genomic deletion [2,16,45,48]. The loss of exon 3 is a crucial element in growth hormone pharmacogenetics, as it alters receptor processing, transport, stability, and binding to other ligands and affects receptor expression or function and signal transduction [49]. The transduction of growth hormone signaling was 30% more efficient through d3GHR homo- or heterodimers than through full-length GHR homodimers. More specifically, the deletion of exon 3 increases the sensitivity of the intracellular JAK-STAT pathway to growth hormone, which in turn increases the transcription of GH target genes [2]. According to an archaic study in 1993, the d3GHR isoform may affect significant changes in physiological function, receptor processing and binding to other ligands [50] and is positively correlated with increased GH responsiveness [3]; although, more recent studies have shown that *GHR* binding is unaffected [2,4,51]. Other research indicated that the deleted region is not close to any binding motifs and that the deletion has no effect on the ligand binding capacity [47]. Additionally, it does not appear that the loss of exon 3 has a significant impact on the intramolecular disulfide bond pattern, indicating that the protein folding is unaffected [2]. However, the newly formed junction between exons 2 and 4 results in the loss of an asparagine-mediated N-glycosylation site, which may have an impact on the glycosylation of the receptor’s extracellular region [52].The effects of increased GH transduction signaling via the d3GHR include decreased feedback on pituitary GH production, lower circulating GH levels, decreased binding affinity to d3GHR-equipped cells, and the restoration of a desirable level of GH action on pertinent targets [53] (Figure 3).

Many studies examined the distribution of the various GHR isoforms among healthy controls and found that, on average, 50% of controls had full-length GHR (fl/fl), 30–40% were heterozygous for the exon 3 deletion (fl/d3), and 10%–20% were homozygous for the exon 3 deletion (d3/d3) [2,16,54,55]. The exon 3 deletion had no impact on receptor binding affinity or internalization and, thus, the molecular mechanisms underlying the higher bioactivity of d3GHR have not yet been clarified [51,56].

## 5. flGHR and d3GHR Genotyping

Genotyping of the d3GHR polymorphism can be achieved via several methods. One of the widely used and prevalent method is utilizing the polymerase chain reaction (PCR) using specific primers [3,8,13,16]. According to Pantel et al., the GHR genotype test can be easily carried out using a PCR assay and the PCR amplification results can be analyzed using agarose gel electrophoresis [16]. The full-length wild type (fl) allele is represented by a PCR fragment of 935 base pairs using the G1 and G3 primers (G3 is an antisense primer located inside exon 3), while the deletion (d3) allele is represented by a PCR fragment of 532 base pairs using the G1 and G2 primers (both primers bracketing exon 3) (GenBank accession number AF155912) [16,54,57].

In addition, the GHR exon 3 deleted/full-length polymorphism can be genotyped quickly and with high throughput using TaqMan SNP genotyping”. rs6873545 is one of several SNPs in the *GHR* that relates to the presence or absence of exon 3. The more prevalent allele encompassing exon 3 correlates with the rs6873545(T) allele, while the less prevalent rs6873545(C) allele correlates with the exon 3-deficient d3GHR allele. In order to identify exon 3 deletion in GHR, rs6873545 has been used as a tag in numerous studies [58,59,60]. In 183 adult GHD patients, Glad et al. tested the TaqMan SNP genotyping using the tagSNP rs6873545 as a marker for the d3/fl polymorphism, and reported that the tagSNP makes it easier to examine the impacts of the d3/fl polymorphism, especially in large cohorts [61].

## 6. Genetic Editing of d3GHR

Numerous studies have been carried out to mimic the deletion of exon 3 in cell lines and animal models utilizing various genome-editing methods to better understand the function and impact of deletion of exon 3 of the GHR and its positive or negative effects. 

In cell lines, researchers transiently cotransfected HEK293 fibroblasts with vectors expressing full-length GHR, d3GHR, or both using the LHRE-luciferase reporter plasmid to investigate whether the GHR is linked to increased responsiveness to growth hormone. When the cells were exposed to different growth hormone concentrations, it was found that d3GHR caused higher transcriptional activity of the reporter construct and responded more strongly to growth hormone stimulation than the full-length GHR [2]. Chiloiro et al. also found that, in comparison to the flGHR isoform, the d3GHR isoform has a higher sensitivity to endogenous and recombinant GH (rhGH) [62].

A further possibility to mimic the deletion of exon 3 is in animal models. Exon 3 of the GHR can be deleted utilizing CRISPR-CAS9 genome editing. Following several backcrosses, the researchers subjected WT and d3GHR mice to AL (ad libitum) feeding and 40% caloric restriction (CR). The findings showed that exon 3 deletion has a protective effect against severe malnutrition; in male mouse livers under CR, the d3GHR causes the expression of female-like genes and the disappearance of sexual dimorphism in weight. The d3GHR males under CR were significantly lighter than the WT males, making them nearly identical in weight to the d3GHR females. In this study, the authors demonstrate how sex and environmental factors can influence the impact of genetic variants [60]. To study the binding properties of d3GHR, scientists have cloned cDNA in a eukaryotic expression vector, transiently expressed the receptor in COS-7 cells, and reported that the receptor lacking exon 3 was expressed on the plasma membrane and was capable of binding hGH with the same high affinity as the flGHR [51].

## 7. Effects of the flGHR and d3GHR Isoforms in Healthy People

Several studies have been conducted to explain the relationship between the d3GHR isoform and birth weight [63], life-span [13], metabolic activity [40], and time of puberty onset [64]. 

Exon 3 deletion is linked to increased height, decreased serum IGF-1, increased GH sensitivity, and longer life expectancy in men. Men homozygous for the presence of exon 3 had an approximately 10-year lifespan increase and were an inch taller than WT isoform carrier individuals [13].

A study on 43 mother–child pairs with normal pregnancies demonstrated that the presence of the d3GHR polymorphism in the fetus is associated with a markedly reduced concentration of hGH-V and IGF-I in maternal serum, a reduced fetal growth rate in the third trimester, and lower birth weight compared to the wild type [65], which can illustrate its association with smaller placental and birth size [63,64]. However, in later developmental stages and in adults, the effect of the d3GHR isoform on organismal growth is reversed. People who carry the d3GHR isoform are more sensitive to GH [13] and there is early onset of puberty in boys when the d3/d3 genotype is compared with the fl/fl genotype [64]. In addition, the d3GHR homozygotes and heterozygotes showed a significantly higher rate of postnatal catch-up growth [5].

In a longitudinal study, 385 healthy individuals from birth to adulthood (ages 23–25) were divided into three groups based on birth size: small for gestational age (SGA n = 130), appropriate for gestational age (AGA n = 162), and large for gestational age (LGA n = 93). The d3/d3 genotype increased in SGA and AGA individuals and decreased in LGA. fl/d3GHR was not related to plasma IGF1 levels in adults, height, weight, metabolic phenotype, or rising cardiovascular risk [66]. In another study, the d3GHR isoform is linked to a lower risk of pre-hypertension in boys but a higher risk of both pre- and hypertension in girls [67].

Yet, in a Saudi population investigation of 230 healthy adults the GHR genotypes were distributed as follows: 39.1% for fl/fl, 44.8% for fl/d3, and 16.1% for d3/d3. They found no discernible differences between the GHR genotypes in height, weight, or body mass index [68]. Further, in healthy Turkish adults (n = 477) 35% fl/fl, 39% fl/d3, and 26% d3/d3; the three GHR genotype groups did not differ in terms of height SDS (standard deviation score) or BMI SDS [69].

## 8. Effects of the flGHR and d3GHR Isoforms on Various Growth Disorders

Numerous studies have shown that GH and GHR deficiencies are the source of a variety of disorders and syndromes, some of which appear in childhood, whereas others have their onset in adolescence and maturity. The responsiveness, timing and efficacy of treatments, GH and IGF-I levels, height, weight, BMI, and other variables were found to be significantly influenced by the d3GHR isoform [2,70,71,72]. Some of the studies found no correlation effects in the growth response and hGH therapy on d3GHR carriers [54], while others had a favorable correlation [2,72], which resulted in two meta-analyses that demonstrated moderate favorable dominant impacts of the d3GHR genotype in the response to GH in d3GHR carriers with short stature [73,74]. Yet, a pharmacogenetic investigation discovered no correlation between growth hormone response and the d3GHR polymorphism in patients with growth hormone deficiency, ISS, and SGA [70]. In conclusion, the results imply that the d3GHR, as examined among flGHR and d3GHR carriers, may be implicated in regulating the effectiveness of GH therapies.

The following sections will discuss how each syndrome is affected by the various GHR genotypes, notably exon 3 deletion.

### 8.1. Growth Hormone Deficiency (GHD)

GHD is a condition in which GH production is reduced or nonexistent. Idiopathic isolated GHD is the most prevalent form of the illness, with an incidence rate of 1:4000 to 1:10,000 [75]. The most common cause of GH deficiency in adult humans is pituitary or peripituitary tumors [76]. In patients suspected of having adult GHD, a diagnosis must be made before replacement therapy with rhGH can be considered [77].

According to numerous studies, children with GHD who lack exon 3 (d3GHR) respond to rhGH more favorably than those who do not [2,78]. GHD patients carrying at least one d3 allele had a noticeably better growth rate in the first year of hGH replacement and attained a taller adult height [72]. After the first two years of recombinant human GH treatment in 181 patients with severe isolated GH deficiency, subjects with the d3/d3GHR isoform significantly outgrew those with the full-length GHR in terms of both height gain and HV SDS score. For both HV SDS and height gain, a dose-dependent effect of the d3GHR isoform was observed [79]. According to a meta-analysis of 15 studies, in GHD, but not in non-GHD, d3GHR is associated with increased baseline height. Additionally, during the first year of rhGH therapy, d3GHR stimulates growth velocity by an additional effect of roughly 0.5 cm. This effect is stronger at lower rhGH doses and at older ages [74].

GH peak level (higher vs lower) and d3GHR (fl/fl vs d3 carriers) combined status was associated with height change over 3 years. d3GHR carriers with lower peak GH had lower growth than subjects with fl/fl. Conversely, d3GHR carriers with higher peak GH had better growth [80]. 

Whereas other results from various studies were inconsistent. Isolated GH deficiency (IGHD) patients who received replacement doses of GH for a year showed that the d3GHR isoform does not affect the response to GH treatment or growth predictions [12]. Caucasian patients who have severe GH deficiency with the fl/d3 or d3/d3 genotype did not require less rhGH therapy to improve their quality of life than those with the fl/fl genotype [81].

### 8.2. Small for Gestational Age (SGA)

Small for gestational age (SGA) fetuses or newborns are those that are smaller than average for their gestational age. This is typically indicated by a weight that is below the 10th percentile [82,83] and affects approximately 3–10% of live births [84,85]. Compared with infants born with a normal length and weight for gestational age, SGA is linked to cardiovascular disease, T2DM, increased risk for insulin resistance, and adults short stature [86,87]. Ninety percent of infants with SGA show catch-up growth, but 10% of them remain diminutive throughout childhood and adolescence [88]. Clinical research revealed that rhGH is a successful treatment for children of short stature who are born with SGA. As a result, children of short stature who are born SGA and who do not achieve catch-up growth by the ages of 2–4 are candidates for rhGH therapy. Children with SGA have been shown to respond differently to rhGH therapy depending on their GH status [84,89]. Numerous studies showed that GH treatment increases adult height in SGA people who did not exhibit catch-up growth [90].

In 536 healthy newborns subjects, SGA (n = 192), LGA (n = 200), and AGA (n = 144), the d3/d3 GHR genotype was found to be twice as common in the AGA and LGA cohorts compared to SGA. However, no significant differences in the frequency distribution of the GHR genotypes between LGA and AGA newborns were found. The information disqualifies the GHR exon 3 deletion polymorphism as a potential genetic cause of LGA pregnancies [91].

Table 1 summarizes the studies that have been published on the relationship between the GHR isoforms and SGA children. Numerous studies support the concept that the d3GHR isoform is linked to a significant increase in compensatory growth in SGA children [7,8] and a better growth response to hGH treatments [2,92]. However, other studies have also supported the lack of a connection between GHR genotypes and patients’ responses to hGH therapy [10,93] (Table 1).

### 8.3. Turner Syndrome

The description of female patients with Turner syndrome (TS) was published by Henry Turner in 1938 [95]. Nevertheless, in 1930, Dr. Ullrich reported a case of an 8-year-old girl with signs of TS; therefore, this syndrome is also called Ullrich–Turner syndrome [96]. This syndrome affects 1 in every 2500 females and is the most prevalent sex chromosome abnormality [97]. Short stature, which affects 98% of patients, is the most prevalent symptom [98]. Between 95% and 100% of TS patients experience growth failure and shorter adult height [99].

The FDA and other regulatory bodies around the world have approved rhGH therapy, which has been shown in numerous studies to increase adult stature in TS with varying effects on final adult height [100,101]. Growth hormone therapy works to increase final adult height, but the treatment benefit varies across studies and is dependent on a variety of factors, including the age when the therapy starts, GH dose, length of therapy, and bone age [9,94,102,103] in addition to the patient’s GHR genotype, which we will discuss in this chapter. As a result, numerous studies have investigated how different GHR genotypes affect the way in which TS patients will respond to GH therapy.

In a study involving 48 TS patients (fl/fl = 24, fl/d3 = 17, and d3/d3 = 7) the d3/d3 group showed a significantly higher total height gain than the fl/fl or the fl/d3 groups (final adult height as compared to their starting height). Additionally, d3GHR homozygous girls display a unique GH responsiveness that controls their weight and led to BMI reduction [9]. Another study found that girls with one or two copies of the d3GHR isoform displayed an increase in height velocity and outgrew their growth prediction during the first year of treatment, but gains in weight, IGF binding protein 3 (IGFBP-3), and IGF-1 were not significantly different [94].

However, a retrospective study of 175 TS patients discovered that the concentrations of HV gain, BMI, IGF-1, and IGFBP-3 between those with and without exon 3 were not significantly different after 1 year of GH therapy [102]. In addition, Turkish TS patients who receive GH treatment for one and two years showed no impact of the GHR exon 3 polymorphism on growth [6].

### 8.4. Prader-Willi Syndrome (PWS)

Prader–Willi syndrome (PWS) is a highly complex and prevalent genetic disorder that occurs in approximately 1 in 15,000–30,000 births. The syndrome is dictated by loss of the paternally inherited region 15 q11-q13 [57,104]. Clinical manifestations evolve with age, with hypotonia and a poor suck leading to failure to thrive during infancy. As the person ages, additional characteristics like short stature, a propensity for food and excessive weight gain, developmental delay, cognitive impairment, and behavioral issues become more noticeable [105]. Individuals with PWS frequently have lower muscle mass and strength as well as severe hypotonia that may increase the risk or tendency for scoliosis, which is determined by standing X-rays that show a skeletal back curve greater than 10° [106]. GH administration is used to treat PWS patients to increase lean body mass and height [104,107].

Possession of the d3 allele confers a protective effect by increasing GH sensitivity in scoliosis over the life-span and improving physiological parameters such as bone density and muscle mass; in fact, those with scoliosis who carried the d3 allele had more gradual increases in weight and height with age than those with the WT genotype [57]. 

In 2011, a study of 74 PWS patients and 100 healthy controls found that patients with at least one d3 allele had significantly higher height SDS and IGF-1 levels than those with flGHR before starting recombinant human growth hormone (rhGH) therapy [108].

In another study, d3GHR isoform carriers experienced a faster 1.5 cm/month increase in height while receiving GH treatment, compared to 0.87 cm/month for those with the WT isoform. Furthermore, in the absence of GH treatment, homozygotes for d3GHR grow faster than both fl/d3 and fl/fl; this result supports the theory that the d3 allele increases sensitivity to GH [109]. According to Schreiner et al., in preterm infants with very low birth weight, postnatal catch-up is significantly more common among those who carry at least one d3 allele than in those who are homozygous for the full-length isoform, defining the d3GHR genotype as a predictor of the postnatal growth pattern [5]. As a result, and in light of the aforementioned studied, the d3GHR isoform has a protective and beneficial effect on PWS patients.

### 8.5. Acromegaly

Acromegaly is a rare disorder caused by excess secretion of GH, usually due to a pituitary tumor [110]. These elevated GH levels lead to an increase in the production of insulin-like growth factor I (IGF-I), mainly in the liver but also in other tissues [111]. Screening for acromegaly is often performed by measuring random GH concentrations and IGF-1. In acromegaly, log GH concentrations are directly correlated with IGF-1 concentrations [112,113]. In active acromegaly patients, the relationship between serum IGF-I and serum GH is gender-specific: females have lower serum IGF-I concentrations than males with the same serum GH level [114]. The distribution of the GHR genotypes (fl/fl, fl/d3, and d3/d3) was examined in numerous studies involving acromegaly patients, and in most studies, roughly half of acromegaly patients carried at least one d3 allele. The prevalence of homozygote d3/d3GHR individuals is 5–20% (Table 2).

Pegvisomant (PEGV) is a brand-new GHR antagonist, which improves symptoms and maintains IGF-1 balance in acromegaly patients over the course of up to 12 weeks of treatment. As a result, PEGV is the best medication for controlling IGF-1 in acromegaly [115]. A study reported that exon 3 deletion predicted a better response to PEGV therapy [11]; for d3GHR carriers, Pegvisomant dosage and treatment time were reduced and IGF-I levels were normalized faster [116]. Other research, however, found that the response to PEGV treatment or determination of the necessary PEGV dose does not differ between GHR genotypes [117].

Numerous studies have examined the connection between various GHR genotypes and several comorbidities linked to patients with active acromegaly, such as hypertension, obesity, T2DM, colonic polyps, heart disease, obstructive sleep apnea syndrome, vertebral fractures (VF), and heart disease, in addition to the effect of each genotype on GH, IGF-I, and BMI levels [44,113,118,119,120,121,122].

**Table 2 ijms-24-13908-t002:** d3GHR polymorphism effects growth parameters in several cohorts of acromegaly patients. Overview of main studies.

Authors	No. of Cases	fl/fl %	fl/d3 %	d3/d3 %	Effects
Schmid, 2007 [113]	44	50	50		d3GHR carriers had lower GH concentrations.
Kamenicky, 2009 [123]	105	51	30	19	No differences in the baseline levels of GH and IGF-1 concentrations.
Pontes, 2020 [121]	88	40	60		No correlation between the presence of d3GHR and increasing frequency of VF, worse BMD, or bone microarchitecture.
Cinar, 2015 [124]	118	60.2	33.9	5.9	No differences in the prevalence of hypertension, type 2 diabetes mellitus, or coronary artery disease.
Montefusco, 2010 [55]	76	55.3	35.5	9.2	d3GHR carriers had a lower BMI index.
Mercado, 2008 [125]	148	45	32	22	d3GHR carriers had higher post-treatment IGF-I concentrations and diabetes was more prevalent in them.
J E Wassenaar, 2009 [126]	86	59	29	7	d3GHR carriers had an increased prevalence of osteoarthritis, dolichocolon, and adenomatous colonic polyps.

As shown by the studies cited above and in Table 2, the effects of the different GHR genotypes on individuals with acromegaly vary considerably. Further, in vivo and in vitro research is required to fully understand how d3GHR impacts the concentrations of GH and IGF-1 as well as other metabolic markers in acromegaly.

## 9. Conclusions and Future Perspectives

According to numerous studies on GHR polymorphism that have been accumulating in recent years, one of the many factors that affect the phenotype of both children and adults with GHD or excess secretion of GH, as well as responsiveness to treatment, is the genotype of the GHR, “flGHR”, or missing exon 3 “d3GHR”. d3GHR, is the first genetic factor that controls how each individual responds to GH treatment. Numerous investigations on the GHR exon 3 genotype in people with various syndromes, including children receiving hGH therapy, girls with Turner syndrome, children born small for gestational age, and patients with acromegaly, corroborated the beneficial effect of the d3GHR isoform on GH actions. In healthy individuals, exon 3 deletion is associated with greater height, a longer life-span, and higher GH sensitivity [13]. However, some studies did not identify any significant differences in height, weight, or other characteristics among the GHR genotypes [66,68].

Several studies on PWS patients who had received GH treatment indicated that those with the d3 allele grew taller faster than those with the fl/fl subtype. This gene confers protection by increasing growth hormone sensitivity in scoliosis over the course of the person’s life and improving physiological features, including bone density and muscle mass. It was also positively connected with spontaneous postnatal growth velocity. Acromegaly patients who receive the GHR antagonist pegvisomant and have the d3GHR isoform typically respond better to treatment than those with the fl/fl GHR isoform. Other research, however, revealed that d3GHR carriers had a higher risk of VF, as well as a high prevalence of obesity and hypertension. In a comparison of girls with d3GHR isoform-positive Turner syndrome and girls with fl/fl GHR, those positive for the d3GHR isoform showed a significantly greater total gain in height. Various studies indicated that d3GHR SGA children carriers grew more spontaneously and responded better to growth hormones, and patients with GHD who lack exon 3 (d3GHR) responded more favorably to rhGH and had higher baseline heights. 

The conflicting findings that have thus far been reported, however, warrant some thought and may be attributed to confounding elements such as small sample sizes, limitations in large prospective studies, variations in experimental design across studies, the individuals’ growth parameters prior to the treatment, sex, and age. In addition, take into consideration that growth rate is a quantitative polygenic variable that is impacted by maternal factors both during pregnancy and afterward. 

We believe that additional studies, particularly sizable prospective studies, should be performed in order to better understand the role of the d3GHR isoform in the various growth disorders and to evaluate the significance of this polymorphisms in GH pharmacogenetics. We suggest that CRISPR-CAS9 or other genome editing technologies to mimic GHR diseases in animal models, followed by the induction of the exon 3 deletion, can be utilized and assist in learning more about the function of the d3GHR isoform in life extension and treatment effectiveness.

## Figures and Tables

**Figure 1 ijms-24-13908-f001:**
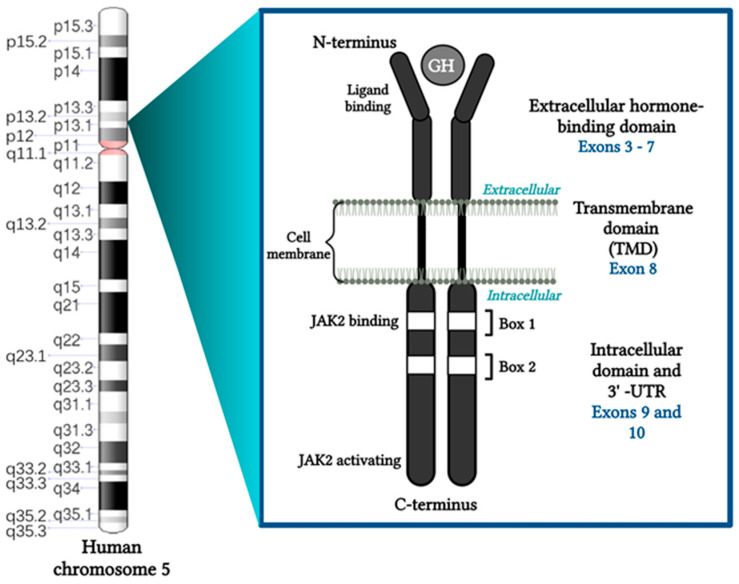
Schematic representation of the GHR. The GHR located on the short arm of chromosome 5, on the 5p13.1-p12 chromosome, consists of an extracellular domain, a transmembrane domain (TMD), and an intracellular domain. Two conserved regions, Box 1 and Box 2, are part of the intracellular domain (created using https://www.biorender.com) (accessed on 1 September 2023).

**Figure 2 ijms-24-13908-f002:**
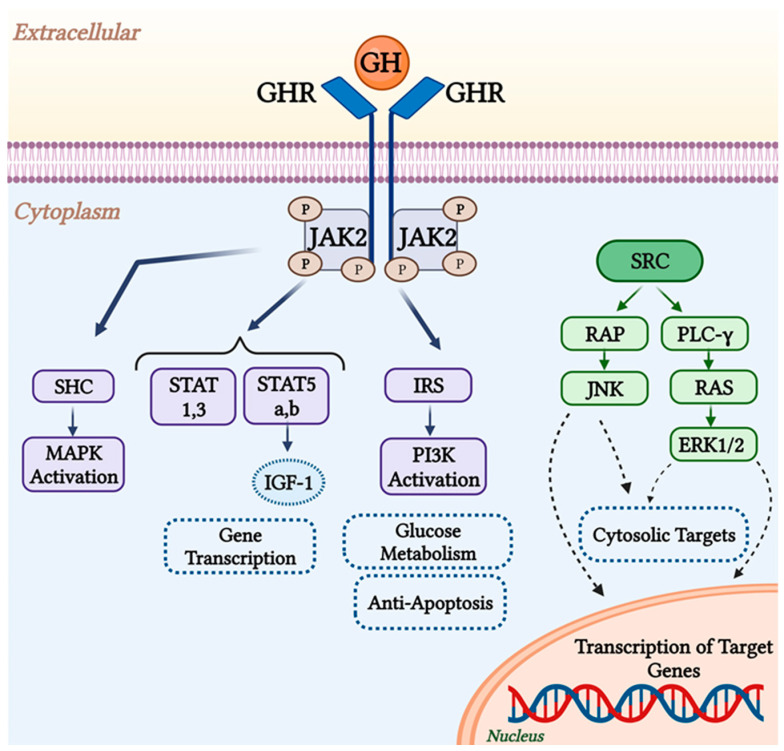
GHR signaling pathways. JAK2 activation by GH initiates certain signaling pathways. SHC is phosphorylated by JAK2, which activates MAPK. JAK2 phosphorylates the transcription factor STAT as well. PI 3′-kinase activation via the phosphorylation of IRS proteins by JAK2 may be crucial for glucose metabolism. The GHR also activates the Src family kinase signaling pathway, which in turn activates JNK and ERK1/2, and regulates gene transcription (created using https://www.biorender.com) (accessed on 1 September 2023).

**Figure 3 ijms-24-13908-f003:**
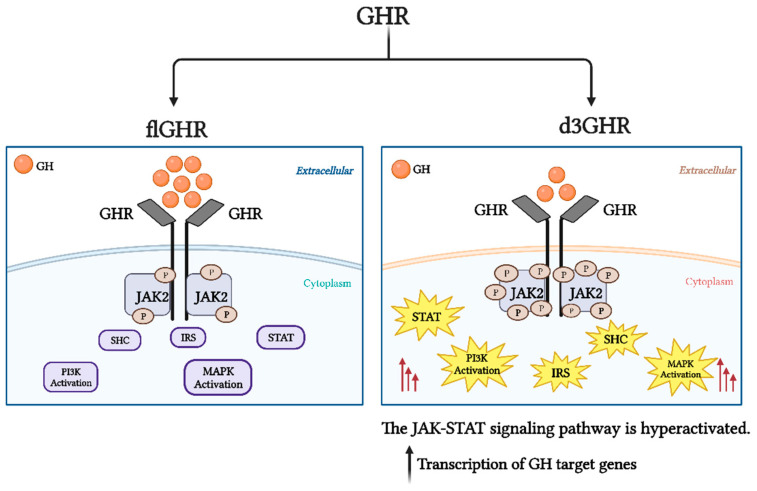
The full-length GH receptor (flGHR) vs. deletion of exon 3 GH receptor (d3GHR) signaling pathways. The sensitivity of the intracellular JAK-STAT pathway to growth hormone is increased in the absence of exon 3, which raises the transcription of GH target genes in comparison to that of the flGHR isoform (created using https://www.biorender.com) (accessed on 27 July 2023).

**Table 1 ijms-24-13908-t001:** Effects of the flGHR and d3GHR isoforms on growth indicators in various cohorts of SGA children.

Authors	No. of Cases	Effects
Wegmann, 2017 [7]	96	d3GHR carriers had increased spontaneous growth, lower IS, and a compensatory increase in glucose, C-peptide, and insulin at baseline.
Garrido, 2021 [8]	65	d3GHR carriers have increased spontaneous growth.
Dos Santos, 2004 [2]	76	d3GHR carriers responded to hGH therapy 1.7 to 2 times better than those who had the flGHR isoform.
Dörr, 2011 [92]	142	d3GHR carriers exhibited a better growth response to growth hormone only in the first year but not in the second year.
Binder, 2006 [94]	60	d3GHR carriers grew significantly more quickly than anticipated. During the first year of rhGH treatment, the average height gain linked to d3GHR was roughly 0.75 cm.
Carrascosa, 2006 [10]	170	Spontaneous growth rate and responsiveness to 66 micro/k·d GH therapy are comparable in flGHR and d3GHR carriers.
Carrascosa, 2008 [93]	60	Both genotypes responded similarly to GH therapy after two years of response (32.1 ± 3.8 microg/kg·d).

## Data Availability

Not applicable.

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
