# Peer review of "The Exon 3-Deleted Growth Hormone Receptor (d3GHR) Polymorphism—A Favorable Backdoor Mechanism for the GHR Function"

_ijms, 2023, doi:10.3390/ijms241813908_

Round 1

Reviewer 1 Report

This review paper by G. Falah et al deals with the of exon 3 deletion polymorphism in GH receptor functions and GH-related pathologies.

General comment :
The present review is very interesting and stimulating.
Therefore it opens the way for other questions (see specific comments) that might be addressed.

Specific comments :

   line 11 : delete gene. GH acts through GHR, not through the GHR gene.

   line 35-41 : This long sentence could be split in two for clarity.

  It is known that only one GH molecule binds two GHR molecules to promote downstream multiple signaling (JAK-STAT, JAK-Shc-MAPK, etc…). Therefore the two GHRs interact with different GH regions of one GH molecule. It is believed that GH initially binds to a single receptor through its so-called “site 1” motif and, subsequently, to the second receptor via a so-called “site 2”.
- Is it known whether presence or absence of exon 3 plays a role in this mechanism ?
- Do d3GHR and wtGHR heterodimerize under GH stimulation? Or are there only d3GHR  homodimers and wtGHR homodimers ?
- Are d3GHR and wtGHR expressed at similar levels at the plasma membrane surface ?

These points should be made clearer.

  The GHR has been observed to be largely localized to the nucleus, under stimulation by GH. This nuclear localization has particularly been observed in proliferating cells including a range of different cancers.
- Is the exon 3 deletion involved in any manner in this GH-induced nuclear localization ?
- And subsequently in cancers ?

Author Response

Dear Reviewer,

Thank you 

Reviewer 2 Report

This review explores the diverse GHR isoforms and summarizes the main pharmacogenetic studies on this ongoing and unresolved topic. I have some comments:

1) This review, which was recently published as a preprint on “https://www.preprints.org/manuscript/202308.1470/v1”. I wonder why the authors chose to submit their work to “preprints.org”, since MDPI seems to offer a similar service. However, if they did this through IJMS, then there is no problem.

2) L14-15: "Many of these studies indicated that the growth response to GH treatment may be affected." This sentence does not convey a clear meaning and needs to be rephrased.

3) L17: Please write the full name of “WT” at its first mention. I understand that it means “wild type”, but other readers may not be familiar with this term. Apply this throughout the whole manuscript.

4) L25-26: Do not put abbreviations in keywords.

5) L51-52: "Growth hormone (GH) is a peptide hormone that is pulsatilely released from the somatotroph cells of the anterior pituitary." Why did you write this sentence in bold? It seems to be a redundant repetition of what you said in the introduction. Repeating sentences can make readers bored and is a weak point of scientific writing.

6) L56-57: "The GH receptor (GHR)" AND "growth hormone receptor (GHR)" You have already used this abbreviation before, so there is no need to write the full name again. In good scientific writing, you should only write the full term at its first mention and then use the abbreviated form throughout the rest of the text.

7) It is very important to italicize all abbreviated names of genes throughout the manuscript. This helps to distinguish between genes and proteins, and eliminates the need to add either “gene” or “protein” after the abbreviated term. This is a standard convention in scientific writing.

8) Fig.1. is not very informative and lacks scientific accuracy. The authors should add a tentative length for each exon and intron and add a scale bar that indicates the length (bp). They should also label exon 1 and draw the corresponding region of the protein instead of writing the name of each domain. This figure should be either improved based on these suggestions or deleted. Also correct 5'UTRs to 5'UTR. It is well known that each gene has one 5'UTR.

9) Cite the software that you used to create Figs. 2 and 3 in the captions. This will help the readers to understand how you generated these diagrams.

10) This article contains many abbreviations, so I suggest that you add a list of abbreviations at the beginning or the end of the article to make it easier for the readers.

11) L84-85: "The GHR also activates the Src family kinase signaling pathway independently of JAK2" This part should be added to Fig. 2.

12) I suggest that you combine Fig. 2 and Fig. 3, since half of Fig. 3 is a redundant repetition of Fig. 2. This will make your figures more concise and clear.

13) L133-135: "Exon 3 is coding for a portion of the extracellular domain, and its deletion results in a loss of 66 base pairs (bp) that code for 22 amino acids in the N-terminal domain (residues 7–28)" This needs a diagram, refer to my previous comment number 8 regarding the protein structure of GHR.

14) L135: "residues 7–28)" Does this mean E2 encodes 6 residues? This could be clearer if you show the amino acid sequence of GHR as suggested before.

15) L136: "alternative splicing of the immature mRNA" Please explain the mechanism of alternative splicing of the immature mRNA in more detail. What factors influence the alternative splicing? Is there a single nucleotide polymorphism (SNP) in the intron-exon junction that causes E3 skipping during splicing? Or is there another reason for this?

16) L147-149: "In addition, the d3GHR isoform facilitates significant changes in physiological function and hormone binding [43] and is positively correlated with increased GH responsiveness [5], although GH receptor binding is unaffected" AND also in L163. This is so confusing, you first said "d3GHR isoform facilitates significant changes in hormone binding" but rather you said "although GH receptor binding is unaffected" this is a contradiction.

17) The authors should conduct a thorough literature review to find relevant publications that investigate whether this mutation affects the conformational structure of the receptor. If there are no such studies, I recommend that the authors perform an in silico study to compare the 3D structures of the WT GHR and the truncated GHR due to the loss of E3.

18) L172: "G1 and G3 primers" and L173 "G1 and G2 primers” Instead of using G1-G3 to refer to the primers, you should specify the location of these primers on GHR and describe the regions that they amplify. 

19) L175-176: “Another common method for genotyping is using a single-nucleotide polymorphism (SNP) which is specific for exon 3 deletion” This is a mistake. SNP is not a method, but a type of mutation that affects only one nucleotide. The mutation you are describing is a deletion mutation or polymorphism, which involves several nucleotides. The methods you can use to detect this mutation include allelic discrimination by qPCR using TaqMan probes. Please correct this error in your manuscript.

20) L251: “In the following chapter” This is not a book.

Minor editing of English language required.

Author Response

Dear Reviewer,

Thank you 

Round 2

Reviewer 1 Report

See comments and suggestions concerning the revised version in the joined Word document below.

Author Response

Dear Reviewer,

Thank you 

Reviewer 2 Report

The authors responded to all my comments and the review was significantly improved.

Author Response

Dear Reviewer,
We are pleased to hear that you found our responses satisfactory and that we have addressed your comments following the major revision.

Thank you